# Depth of Neuromuscular Block Is Not Associated with Abdominal Wall Distention or Surgical Conditions during Gynecologic Laparoscopic Operations. A Prospective Trial

**DOI:** 10.3390/jcm9041078

**Published:** 2020-04-10

**Authors:** Stefan Soltesz, Alexander Mathes, Michael Anapolski, Karl Guenter Noé

**Affiliations:** 1Department of Anesthesia and Intensive Care, Rheinland Klinikum Dormagen, 41540 Dormagen, Germany; 2Department of Anesthesia and Intensive Care Medicine, University Hospital of Cologne, 50924 Cologne, Germany; 3Department Ob/Gyn, University of Witten-Herdecke, Rheinland Klinikum Dormagen, 41540 Dormagen, Germany; michael.anapolski@kkh-ne.de (M.A.); karl-guenter.noe@kkh-ne.de (K.G.N.)

**Keywords:** deep neuromuscular block, surgical site conditions, laparoscopic surgery

## Abstract

The influence of the degree of a neuromuscular block (NMB) on surgical operating conditions during laparoscopic surgery is debated controversially. The extent of abdominal distension during the time course of the NMB was assessed as a new measurement tool. In 60 patients scheduled for gynecologic laparoscopic surgery, the increase of the abdominal wall length induced by the capnoperitoneum was measured at 5 degrees of the NMB: intense NMB—post-tetanic count (PTC) = 0; deep NMB—train-of-four count (TOF) = 0 and PTC = 1–5; medium NMB—PTC > 5 and TOF = 0–1; shallow NMB—TOF > 1; full recovery—train-of-four ratio TOFR > 90%. Simultaneously, the quality of operating conditions was assessed with a standardized rating scale (SRS) reaching from 1 (extremely poor conditions) to 5 (excellent conditions). Fifty patients could be included in the analysis. The abdominal wall length increased by 10–13 mm induced by the capnoperitoneum. SRS was higher during intense NMB (4.7 ± 0.5) vs. full recovery (4.5 ± 0.5) (mean ± SD; *p* = 0.025). Generally, an intense NMB did not increase abdominal wall length induced by capnoperitoneum. Additionally, its influence on the quality of surgical operating conditions seems to be of minor clinical relevance.

## 1. Introduction

During laparoscopic surgery, insufflation of carbon dioxide (CO_2_) creates a space enabling the surgeon to perform the operation. Many factors contribute to the volume of this space, such as the anatomy of the patients (height, weight, adhesions, etc.), as well as intra-abdominal pressure, positioning of the patient, or anesthetic depth.

Many studies examined the influence of the neuromuscular block (NMB) on the quality of operating conditions [1,2,3,4,5,6,7,8,9,10]. Therefore, intraabdominal measurements, e.g., of abdominal volume, were performed [2,5,6], and several scores were developed [4,8,11]. However, assessing operating conditions objectively proved to be a difficult task, which might be a reason for differing results with respect to the degree of NMB required.

Therefore, the objective of this study was to assess the association of the NMB on abdominal distension and operating conditions. To obtain a reliable parameter and a new measurement tool for this purpose, the degree of abdominal distension during the time course of the NMB was assessed and compared with a standardized score (surgical rating score, SRS) [8]. The primary outcome parameter was the association between the NMB and the increase of the abdominal wall length in longitudinal direction induced by the capnoperitoneum. We hypothesized that an increasing degree of the NMB would correspond to an increase in length of the abdominal wall length.

Assuming that the distension occurs in all three dimensions, an increase of 10% in longitudinal direction would correspond to an increase of 30% in volume [2].

A second outcome parameter was to evaluate if sufficient operating conditions assessed by the SRS correlated with the degree of the NMB.

## 2. Experimental Section

### 2.1. Patients

After obtaining approval from the local ethics committee (Universität Witten-Herdecke, protocol number 220/2017, date of approval: 24 January 2018), and written informed consent, 60 female patients aged 18 years or older, American Society of Anesthesiologists (ASA) Physical Status 1–2, and scheduled for elective gynecologic laparoscopic surgery, such as hysterectomy, pectopexy, and colposuspension, were included in this prospective, blinded study (ClinicalTrials.gov identifier NCT03605628). Exclusion Criteria were an anticipated difficult airway, increased risk for pulmonary aspiration, pregnancy, impaired liver or kidney function, neuromuscular disease, or chronic intake of drugs known to influence neuromuscular blockade. Acute infections are able to alter the sensitivity to neuromuscular blocking drugs: these patients require higher doses. Additionally, the risk of perioperative complications is increased. Therefore, patients with fever, elevated leucocyte counts, or C-reactive protein, were excluded from the study.

### 2.2. Anesthesia

Anesthesia was induced with propofol 2–3 mg kg^−1^ and sufentanil 0.3 µg kg^−1^ and was maintained as a total intravenous anesthesia with propofol 3–5 mg kg h^−1^ and remifentanil 0.1–0.3 µg kg min^−1^ (manually programmed infusion devices, infusion rates based on actual body weight). The patients’ lungs were ventilated to normocapnia, defined as end-tidal carbon dioxide of 35–45 mmHg, via a laryngeal mask airway using a standardized lung-protective strategy (tidal volume 6–8 mL kg^−1^), with a positive end expiratory pressure of 3 cm H_2_O and an inspiratory pressure of 10–15 cm H_2_O.

Neuromuscular transmission was measured by assessment of the post-tetanic count (PTC), train-of-four count (TOF), and the train-of-four ratio (TOFR) using acceleromyography (TOF Watch SX™, Essex Pharma GmbH, Munich, Germany) at the right adductor pollicis muscle with transcutaneous Ag/AgCl electrodes (electrocardiogram electrodes; Ambu Inc., Columbia, MD, USA). To establish a control twitch height value of 100%, the acceleromyograph was calibrated to a delivered supramaximal TOF stimulus (0.2 Hz every 15 s, duration 0.1 ms). The TOF Watch SX possesses a calibration function which automatically determines the individual supramaximal stimulation current (up to a maximum current of 60 mA). The maximal acceleromygraphic response is automatically stored and serves as a reference control value for all subsequent measurements [12]. The first of the four twitch height responses was considered to be T1, and the TOF ratio was the ratio of the fourth twitch (T4) height response compared to T1. During the first min of stimulation, the acceleromyographic signal frequently drifts because the electric current leads to changes of the Ag/Cl electrode impedance. Therefore, after a 10 min period of stabilization, it was recalibrated. Then, patients received rocuronium at 0.6 mg kg^−1^ ideal body weight (2 × ED 95%). Afterward, the degree of the NMB was assessed using TOF stimulation every 15 s by acceleromyography. All acceleromyographic data were recorded continuously on a laptop connected to the acceleromyograph. All patients received a warming device (Bair Hugger Model 505 convective warming unit, Augustine Medical. Inc., Eden Prairie, MN, USA) over the head, thorax, and right arm, and the core temperature and the surface temperature of the arm were monitored continuously. If the initial dose of rocuronium did not lead to a PTC of 0, additional amounts were administered until this goal was achieved. The laryngeal mask airway was exchanged by an endotracheal tube after onset of the NMB.

The degree of the NMB was defined as follows [13,14,15]: intense NMB—PTC = 0; deep NMB—TOF = 0 and PTC = 1–5; medium NMB—PTC > 5 and TOF = 0–1; shallow—TOF > 1; full recovery—TOFR > 90%. If an intense NMB was not reached with the initial dose of rocuronium, the anesthesiologist administered additional doses until a PTC = 0 could be recorded at least once for each patient. Thus, the order of occurrence of the different degrees of the NMB could vary between patients. This approach ensured that the surgeon performing the assessment of the surgical conditions could not anticipate continuously decreasing degrees of the NMB during the time course of the measurements.

To exclude insufficient operating conditions due to inadequate depth of anesthesia, all patients were monitored with a bispectral index (BIS, Anandic Medical Systems AG, Feuerthalen, Switzerland). Propofol and remifentanil infusion rates were adjusted by the attending anesthesiologist in order to reach a BIS value from 40 to 50. In case of insufficient anesthesia (sudden rise of BIS, heart rate, or blood pressure), additional bolus injections of sufentanil were administered, and the propofol and remifentanil infusion rate could be increased. If the surgeon rated the operation conditions as insufficient (SRS 1–3), the patients received 10 mg rocuronium intravenously.

All patients were positioned in Trendelenburg position (30°), and the capnoperitoneum was adjusted and automatically maintained to 12 mm Hg in all cases.

In case of incomplete recovery from the NMB at the end of the surgery, patients received neostigmine (1–2 mg) and atropine (0.5–1 mg) intravenously. Extubation was performed when full recovery was observed (TOFR > 0.9).

### 2.3. SRS

The surgeon assessed the quality of the surgical conditions by means of a standardized score: the Surgical Rating Score (SRS), introduced by Martini et al. and commonly used for this purpose [8]. It has been validated for laparoscopic gynecologic surgery [4,6,16], and its scale ranges from 1–5, defined as follows:

1: Extremely poor conditions: the surgeon is unable to work because of coughing or because of the inability to obtain a visible laparoscopic field because of inadequate muscle relaxation. Additional neuromuscular blocking agents must be given.

2: Poor conditions: there is a visible laparoscopic field, but the surgeon is severely hampered by inadequate muscle relaxation with continuous muscle contractions, movements, or both with the hazard of tissue damage. Additional neuromuscular blocking agents must be given.

3: Acceptable conditions: there is a wide visible laparoscopic field, but muscle contractions, movements, or both occur regularly, causing some interference with the surgeon’s work. There is a need for additional neuromuscular blocking agents to prevent deterioration.

4: Good conditions: there is a wide laparoscopic working field with sporadic muscle contractions, movements, or both. There is no immediate need for additional neuromuscular blocking agents unless there is a fear of deterioration.

5: Optimal conditions: there is a wide visible laparoscopic working field without any movement or contractions. There is no need for additional neuromuscular blocking agents.

Assessment of the quality of surgical conditions was performed by the same surgeon without information with regard to the actual depth of the NMB or the drugs administered for improvement of the block. Measurements were performed every 15 min or in case of deterioration of the SRS. Additionally, the SRS was assessed as soon as possible after one of the 5 degrees of the NMB mentioned above were reached for the first time during the operation: PTC = 0, PTC from 1–5, TOF 0–1, TOF 2–4, and TOFR > 90% (Figure 1). The surgeon was not informed about any of the results of the acceleromyography before all patients had been enrolled.

### 2.4. Abdominal Wall Length

Following calibration of the acceleromyography and endotracheal intubation, as described above, all patients were positioned in the Trendelenburg position. Afterwards, the measurement of change in abdominal wall length (mm) during NMB was performed before the operation with a measuring tape placed on the abdominal wall in the longitudinal direction on the left side of the umbilicus (Figure 2). The caudal end of the tape was fixed with plaster, while the cranial end was free to move. A distance of 100 mm was marked on the skin of the patient. Afterwards, the capnoperitoneum was established and the change of the abdominal wall length was measured at the 5 time points mentioned above, and, additionally, in case of sudden deterioration of the surgical conditions.

### 2.5. Statistics

Statistical analysis was performed using the software package Sigma Plot 12.3 for Windows (Systat Software Inc., Chicago, IL, USA). Mean and standard deviation were calculated for patients’ characteristics and initial rocuronium dose. One-way repeated measures analysis of variance was used to detect differences between the time points for all other data. To isolate the time points that differed from the others, all pairwise multiple comparison procedures were performed with the Holm–Sidak method. The sample size was calculated prior to the recruitment. The measurement of the abdominal wall length has not been published previously; therefore, we were not able to rely on data of previous investigations for sample size calculations. Two similar studies measuring the skin-sacral promontory distance during capnoperitoneum, calculated with 5 mm difference (SD 5 mm) and 12 patients [2], and 20 mm difference (SD 16 mm) and 14 patients, respectively [6]. Torensma et al. calculated, with 20 patients, a difference of 0.5 points (SD = 0.4), with regard to the SRS [10].

However, since our measurements were new, our sample size calculation was as follows: To find a difference of 10 mm of abdominal wall length (SD 20 mm) between the measurement during an intense NMB as baseline and the other time points with a power of 0.9 and *p* < 0.05, 45 patients were required. The drop-out rate was generously calculated with 15 patients because we could not rely on previous data.

## 3. Results

Three patients were excluded because of technical problems with the acceleromyography. Additionally, in seven cases, the duration of the operation was too short to obtain all the scheduled measurements during the surgical procedure. These patients were excluded from analysis and received neostigmine/atropine. The tracheal tube was removed after complete neuromuscular recovery (TOFR > 0.9) several minutes after the end of the surgical procedure. Thus, 50 patients could be included in the analysis. Details are presented in Figure 3 and Table 1.

Twelve patients required additional doses of rocuronium to reach a PTC of 0. Propofol was administered with 3–4 mg kg h^−1^, remifentanil with 0.12–0.14 µg kg min^−1^, and differences between the time points could not be observed. Because of inadequate analgesia, sufentanil bolus injections had to be performed at all degrees of the NMB. Because of coughing or movements resulting in inadequate surgical conditions, additional rocuronium injections were required in five patients while the TOF was 0–1, and in one patient while the TOF was > 1 (Table 2).

With this regimen, the goal of similar anesthetic depth was achieved, as can be seen from the BIS values in Table 3. Slightly higher arterial pressures compared to PTC = 0 were recorded at the measurement points PTC 1–5, TOF 0–1 and TOF > 1, respectively. The heart rate was higher at TOF 0–1, compared to TOFR > 90% (Table 3).

After implementation of the capnoperitoneum, abdominal wall length increased from 10.5 (4.5) mm to 11.5 (4.6) mm compared to baseline. However, differences with respect to the degree of the NMB were not detected (Table 4).

SRS scores were similar during the time course of the NMB. Differences occurred merely between the assessments at PTC = 0 and TOFR > 90%, respectively (Table 5).

## 4. Discussion

The principal finding of this investigation was that an intense NMB did not increase abdominal wall length induced by capnoperitoneum when compared to lower degrees of an NMB, or even no NMB at all.

Moreover, another result of clinical relevance was that an intense NMB did not improve surgical conditions compared to a light block during gynecologic laparoscopic surgery. At first glance, this result seems to be surprising. Nevertheless, it is in line with the ongoing controversial debate on neuromuscular blockade to improve surgical conditions. Whereas many publications present significantly better operating conditions during deep NMB, compared to a light block [3,4,8,14,17,18,19], several other authors found no differences [1,15,20]. Park et al. describe inconsistent results and state that “further studies are required to address the heterogeneity and power shortage demonstrated by the trial sequential analysis”, in a previously published review [21]. Most of these studies were performed with the same five-point SRS used in our investigation. Therefore, this tool has been validated in many investigations [16].

In our study, SRS values were similar throughout the duration of the operation, irrespective of the degree of the NMB. Differences could only be observed between intense NMB and no block at all. However, although this was statistically significant, an SRS score difference of 0.2 is likely to be of minor clinical relevance.

It has been shown that BIS values might be reduced by the NMB without anesthesia and that therefore the BIS might not be eligible as a measure of adequate anesthetic depth during anesthesia, if neuromuscular blocking agents are administered [22]. In spite of this, the present study could not observe any associations between BIS and depth of the NMB. BIS and hemodynamic parameters were similar during all states of the NMB. Additionally, the required doses of anesthetics and analgesics did not differ between the time points of measurements. Thus, it can be assumed that depth of anesthesia was equal during the time course of the measurements.

Several other authors measured changes of the intra-abdominal space induced by the capnoperitoneum with a similar approach: Lindekaer et al. were able to demonstrate a change of the distance between the promontory and the skin, induced by different intraabdominal pressures and by different degrees of the NMB, in a small study including 15 patients scheduled for elective gynecologic surgery [5]. Additionally, Barrio et al. evaluated the skin-sacral promontory distance in 35 gynecologic patients during different degrees of an NMB. They observed an increase during deep NMB compared to light block [2]. Only one investigation correlated measurements of the abdominal space with SRS: Madsen et al. assessed the skin-promontory distance and changes in SRS during deep NMB in 14 patients. They found a significant increase of the abdominal space, together with a higher SRS, during deep NMB [6].

In spite of the fact that we included substantially more patients than in the studies mentioned above, we were not able to detect differences of the increase in abdominal wall length induced by different degrees of the NMB. This result leads to the question if the measurement of the abdominal wall length generally is a suitable tool in order to assess the NMB adequately: our method has not been published before; therefore data providing information with regard to its reliability or validity are not available. Unfortunately, we are not able to provide evidence that our method reflects the NMB adequately, based on our data alone. Possibly, an adequate depth of anesthesia reduces the tension of the abdominal wall similarly to an NMB.

This study was strengthened by the fact that the surgeon was blinded to the acceleromyograph measurements and thereby unaware of the depth of NMB when assessing operating conditions.

Another advantage was the assessment of five different degrees of the NMB instead of only two, as in many other investigations. By this approach, we were able to evaluate all degrees of the NMB, from an intense block (PTC = 0) to full recovery, without inter-individual differences. Additionally, the definition of deep or light NMB varies slightly between different investigations, sometimes making it difficult to compare the results of different studies [14,15,23]. By measuring more than two different degrees, we avoided this problem.

A methodological problem of the study was the potential bias of surgeons in their assessment of SRS: if a surgeon requests an increase of the NMB because of inadequate surgical site conditions, he/she might rate the following SRS higher than it actually is, since he/she expects an improved SRS score because of the rocuronium injection. The injection of a placebo instead of rocuronium would have been a possibility to minimize this problem. However, this was an observational study, and therefore, we did not want to compromise the results of the surgery by this measure. Moreover, the total amount of SRS measurements was substantially higher than the number of additional rocuronium injections: one assessment every 15 min during a time interval of 1–2 h, one assessment for each change of degree of the NMB, and additional assessments in case of a deterioration of the SRS, were required. Thus, with regard to the collective of 50 patients, several hundred SRS assessments had to be performed by the surgeon. By contrast, only six additional rocuronium injections were necessary. Therefore, the influence of six assessments following additional rocuronium injections might be of minor effect.

In many cases, rocuronium had not to be administrated even when no block at all was present. Neuromuscular blocking agents offer the advantage of better intubation conditions with less tracheal injury [24,25]. However, repeated intraoperative injections provide no further benefit with regard to better operating conditions, but might instead increase the risk of postoperative complications, such as respiratory distress or pulmonary aspiration, even if a TOFR > 90% is achieved at the end of surgery [26].

## 5. Conclusions

In our study, an intense NMB did not increase abdominal wall length induced by capnoperitoneum. In gynecologic laparoscopic operations, intense NMB neither increases abdominal wall length, nor the quality of surgical operating conditions.

## Figures and Tables

**Figure 1 jcm-09-01078-f001:**
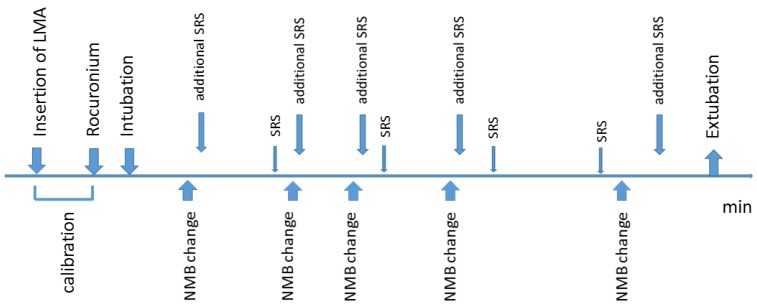
Timeline of the observations. Acceleromyography was performed every 15 s. Surgical rating score (SRS) was assessed (i) every 15 min, (ii) when the degree of the neuromuscular block (NMB) changed, and (iii) in case of deterioration of the surgical conditions, requiring injection of rocuronium. LMA: laryngeal mask airway; SRS: regular assessment of SRS every 15 min; NMB change: change of the degree of the NMB; additional SRS: assessment of SRS in case of change of the degree of the NMB.

**Figure 2 jcm-09-01078-f002:**
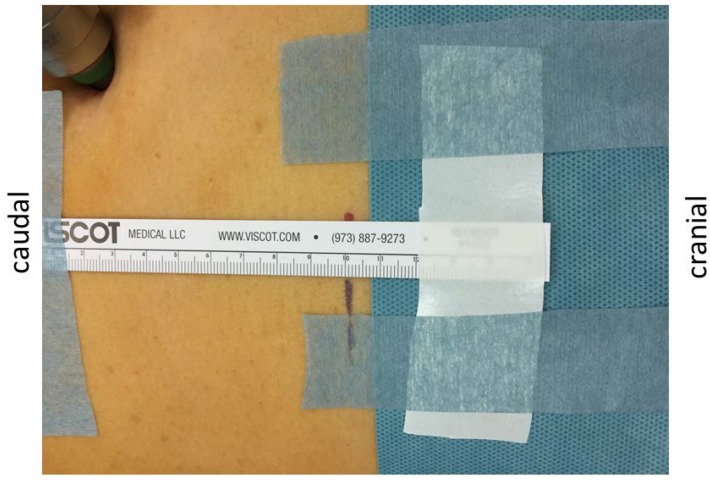
Measurement of the abdominal wall length. The flexible tape is fixed at the caudal end, while it is free to move at the cranial side. A distance of 100 mm from a fixed caudal point (not visible in the figure) is marked cranially (black line).

**Figure 3 jcm-09-01078-f003:**
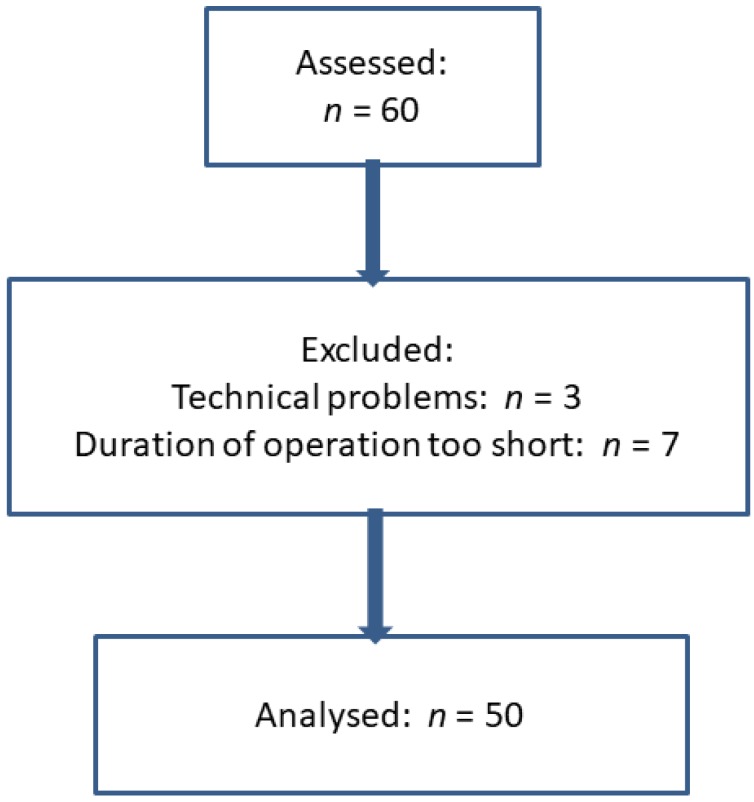
Consort flow diagram of the patients.

**Table 1 jcm-09-01078-t001:** Patient characteristics (*n* = 50) and initial rocuronium administration.

Variables:	Age (years)	Height (cm)	Weight (kg)	BMI (kg/cm^2^)	Rocuronium (mg) Initial Dose	Rocuronium (mg) Repetition
Mean (SD)	50.9 (13.4)	167.4 (5.4)	70.8 (13.9)	25.2 (4.5)	37.4 (3.8)	17.9 (7.2)

BMI: Body mass index; SD: standard deviation. Rocuronium initial dose: 0.6 mg kg^−1^ (calculated according to ideal body weight), rocuronium repetition: additional dose required to reach a post-tetanic count of 0 (*n* = 12). PTC: post-tetanic count; TOF: train-of-four count; TOFR: train-of-four ratio.

**Table 2 jcm-09-01078-t002:** Administration of propofol, analgesics, and rocuronium.

Drugs	PTC = 0	PTC = 1–5	TOF 0–1	TOF > 1	TOFR > 90%
Propofol infusion: mg kg h^−1^ (mean (SD))	3.7 (0.7)	3.8 (0.8)	3.6 (0.8)	3.8 (1.0)	3.8 (0.9)
Remifentanil infusion: µg kg min^−1^ (mean (SD))	0.12 (0.03)	0.13 (0.03)	0.13 (0.03)	0.14 (0.03)	0.14 (0.04)
Sufentanil bolus: number	12	14	3	12	3
Sufentanil bolus: mg (mean (SD))	15.0 (7.1)	11.1 (5.6)	13.3 (5.8)	10.0 (4.3)	8.3 (2.9)
Rocuronium bolus: number	0	0	5	1	0
Rocuronium bolus: mg (mean (SD))	-	-	13 (4.5)	10	-

SD: standard deviation. Number: the cumulative numbers of injections were counted during the following time intervals: from the time point when the degree of the neuromuscular block was recorded for the first time, until the next degree of the neuromuscular block was reached, or until extubation, respectively (for TOFR > 90%). PTC: post-tetanic count; TOF: train-of-four count; TOFR: train-of-four ratio.

**Table 3 jcm-09-01078-t003:** Bispectral index and hemodynamic parameters.

Variables:	PTC = 0	PTC = 1–5	TOF 0–1	TOF > 1	TOFR > 90%	*p*-Value
BIS	41.6 (5.5)	41.9 (6.3)	41.3 (5.0)	42.8 (5.5)	43.5 (4.6)	*n*.s.
BP syst (mm Hg)	96.8 (18.2)	105.6 (21.1)	110.1 (18.0) *	109.4 (16.6) *	102.4 (15.7)	*: *p* < 0.001 vs. PTC = 0
BP diast (mm Hg)	56.6 (11.7)	63.4 (13.7) *	67.8 (10.1) * ^+^	65.6 (10.2) * ^+^	60.0 (7.7)	*: *p* < 0.001 vs. PTC = 0 ^+^: *p* <0.008 vs. TOFR > 90%
HR	56.1 (8.9)	56.9 (8.1)	58.3 (9.6) ^#^	57.4 (8.8)	55.9 (8.8)	^#^: *p* < 0.05 vs. TOFR > 90%

BIS: bispectral index; BP: blood pressure; HR: heart rate. Values are presented as mean (SD). PTC: post-tetanic count; TOF: train-of-four count; TOFR: train-of-four ratio. *: *p* < 0.001; ^+^: *p* < 0.008; ^#^: *p* < 0.05.

**Table 4 jcm-09-01078-t004:** Measurement of the increase in abdominal wall length.

Abdominal Wall Length	PTC = 0	PTC = 1–5	TOF 0–1	TOF > 1	TOFR > 90%	*p*-Value
Increase in mm: mean (SD)	11.5 (4.6)	11.5 (4.9)	10.5 (4.5)	11.1 (5.9)	10.6 (5.2)	0.17

Increase of abdominal wall length: a distance of 100 mm was marked on the abdominal wall after induction of anesthesia. This distance was regarded as baseline. Afterward, the capnoperitoneum was established and the distance was measured repeatedly during different degrees of the NMB. The difference between baseline and the consecutive measurements was defined as the increase in abdominal wall length. PTC: post-tetanic count; TOF: train-of-four count; TOFR: train-of-four ratio.

**Table 5 jcm-09-01078-t005:** Surgical rating scale (SRS).

SRS	PTC = 0	PTC = 1–5	TOF 0–1	TOF > 1	TOFR > 90%	*p*-Value
SRS (mean (SD))	4.7 (0.5)	4.6 (0.5)	4.6 (0.5)	4.6 (0.6)	4.5 (0.5) *	*: *p* = 0.025 vs. PTC = 0

PTC: post-tetanic count; TOF: train-of-four count; TOFR: train-of-four ratio. *: *p* = 0.025.

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
