# Peer review of "Depth of Neuromuscular Block Is Not Associated with Abdominal Wall Distention or Surgical Conditions during Gynecologic Laparoscopic Operations. A Prospective Trial"

_jcm, 2020, doi:10.3390/jcm9041078_

Round 1

Reviewer 1 Report

Second round of comments for authors.

Thank you very much to the editor for allowing me to review this interesting paper once again. Thank you very much to the authors for improving the manuscript significantly.

I will address some comments that the authors provided in their revision letter point by point and add some new comments as well. Before I can recommend this paper for publication, this must be attended to be the authors.

General

I would suggest providing a simple timeline depicting the time points of anesthetic interventions and the different measurements.

  • A timeline has been inserted (figure 1 of the revised manuscript. The other figures have been renumbered).
  • This is appreciated. Furthermore:
    • Please refer to figure 1 a “Timeline of observations”, since “interventions” make it sound like a randomised trial has been performed.
    • Please explain the abbreviation LMA in the figure legend.
    • Please add graphics about point i) and ii) from the legend to the actual timeline.
    • The fact that “the order of occurrence of the different degrees of the NMB could vary between patients” (line 96-97) is not reflected in the timeline. The timeline makes it seem that the different degrees of the NMB appear in a fixed order. Please attend to this.

  1. Introduction
  2. The goal of present study was to evaluate the influence of the NMB on abdominal distention. This is stated very clearly but should be changed from ‘influence’ to ‘association between’ as this is an observational study that cannot estimate causality.
  • The term has been changed accordingly (line 41 and 45).
  • Please change this in the title as well.
  1. In the discussion, the authors explain that a 10% increase in abdominal wall length corresponds to a 30% increase in abdominal volume. I believe that this is the sole argument for being interested in abdominal wall length during surgery and that this should therefore be presented in the introduction with the appropriate reference.
  • The recommendation of reviewer 1 has been added to the introduction (line 48-49).
  • This is appreciated. Please remove the phrase from the discussion (line 260-261), since it is now a repetition. Also, the sentence prior to this (line 259-260) seems out of place and not in coherence with the text preceding it and could be removed as well: “At first glance, a difference of 10-13 mm might be considered to be of minor importance.”
  1. In line 32-34, the authors write: “Many factors contribute to the volume of this space, such as the anatomy of the patients (height, weight, adhesions etc), as well as intraabdominal pressure, positioning of the patient, anesthetic depth, or degree of a neuromuscular block (NMB).” I would recommend the last part (“or degree of a neuromuscular block (NMB)”) removed as this is not yet proven but instead is exactly what the authors a trying to prove according to line 40-41: “Therefore, the objective of this study was to assess the association of the NMB on abdominal distension and operating conditions.”

  1. Method

2.1 Patients

  1. There is some disagreement between the eligibility criteria in the article and in the protocol. The protocol states, that only patients weighing 50-90 kg were included, and this is not stated in the article. Was this exclusion criteria upheld? If so, why was it applied?
  • Patients weighing >90 kg were also included. We changed the inclusion criteria, because a restriction of the body weight would not have changed the results or the significance of the study.
  • Would the authors please include this change in inclusion criteria as a protocol amendment as soon as possible? In future meta-analyses or systematic reviews including present article, this will be of value and help deem present article less susceptible to bias.
  1. Why were patients with acute infections excluded? Did it apply to minor infections such as sinusitis and uncomplicated UTI?
  • Asymptomatic patients without fever and without elevated leucocyte counts or CRP were included into the study. However, acute infections with elevated CRP or PCT are able to alter the sensitivity to neuromuscular blocking drugs: these patients require higher doses. Additionally, the risk of perioperative complications is increased. These were the reasons why we did not include patients with severe infections. The information now is provided (line 61).
  • Thank you for the explanation. Will the authors please include this information about sensitivity to NMB and infection to the text?

2.2 Anesthesia

  1. Was the administration of anesthesia and analgesia (propofol, sufentanil and remifentanil) based on ideal body weight or total body weight? If it was based on ideal body weight, the patients with high BMI could have been given relatively low doses which might explain the subgroup effect of the depth of the NMB in these patients.
  • Propofol, sufentanil, and remifentanil were administered based on actual body weight. Only rocuronium was administered based on ideal body weight.
  • Will the authors please add this information to the text?
  1. The authors write in line 93-94: “The degree of the NMB was defined as follows: intense NMB: PTC=0; deep NMB: TOF=0 and PTC=1-5; moderate NMB: PTC>5 and TOF=0-1; light: TOF>1; full recovery: TOFR>90%.” This definition of NMB degrees is not in coherence with the one in line 18-20. Please attend to this.

2.3 SRS

  1. The authors highlight the fact that the surgeon did not know how deep the neuromuscular block was when assessing SRS (line 231-235). But isn’t the surgeon aware of the fact that extreme NMB (PTC = 0) was the goal of the initial administration of rocuronium? If so, the surgeon was actually only truly “blinded” to the 12 patients that didn’t reach extreme NMB during the initial block. Additionally, Table 1 legend says 12 patients, but the text says 13 patients needed another dose of rocuronium to reach PCT=0.
  • The correct number of 12 patients now is presented in table 1 and the text (line 197).
  • The number is incorrect (13) in line 271.

  • Also, the text says “complete” block (line 272) but PTC=0 is defined as “intense”.
  • As a matter of fact, the majority of patients had a PTC= 0 at the first assessment. However, the surgeon was not aware of this. The proportions could have been completely different. The surgeon never was informed about the results of the acceleromyography during or after the operation. The surgeon was not informed about any of the results of the acceleromyography before all patients had been enrolled. This information has been added (line 140-141).
  • It the surgeon was in fact blinded to all acceleromyograph measurements, then why highlight the fact that 12 patients didn’t develop an intense block to begin with (line 271-275)? I would recommend removing this paragraph entirely and replacing it with something like: “This study was strengthened by the fact that the surgeon was blinded to the acceleromyograph measurements and thereby unaware of the depth of NMB when assessing operating conditions.” Please see comment 4.f for additional changes to this paragraph.

  • I wonder why some patients (12) didn’t reach PTC=0 at first measurement, when PTC=0 was already reached at baseline abdominal wall length measurement before capnoperitoneum was established (according to table 4 legend) and before surgery began. Could you explain this please?

  1. In line 134-136, the authors write: “Assessment of the quality of surgical conditions was performed by the same surgeon without information with regard to the actual depth of the NMB, or the drugs administered for improvement of the block.” How is the last author of this manuscript (who is also an experienced surgeon) unaware of the drugs administered (rocuronium) for improvement of the block? I would expect that he is familiar with the trial protocol (which should specify what agents are utilized in said trial). I would also expect that he might be likely to be familiar with what is probably the standard neuromuscular blocking agent at the surgical ward.

2.4 Abdominal wall length

  1. It remains somewhat unclear to me how the change in the abdominal wall length was measured. Figure 1 does not reveal anatomical marks and I am not able to identify what part of the abdomen is being measured. Is the caudal end of the patient found in the left side of the picture? Is the measuring tape under the white tape free to move or is it fixed? The tape looks rigid like a ruler, but it might not be. Does the black line mark a distance of 100 mm from the tape on the left side of picture? I would suggest providing more information about the figure and/or add a schematic diagram of the set up.
  • Figure 1 (now figure 2 in the revised manuscript) has been improved (line 157-161):
    o Caudal and cranial directions are marked now.
    o The tape is free to move under the white tape on the cranial side while it is fixed caudally.
    o The tape is flexible.
    o The black line marks a distance of 100 mm from the left (caudal) end of the tape.
  • Thank you for the clarification. According to the figure, the black line marks only a distance of 850 mm from the caudal end – if the “caudal end” is defined as the blue tape near the 1,5 cm mark. In reality, I’m sure that the authors measured and marked a proper distance of 100 mm, but the picture doesn’t support this. Perhaps change the figure legend to: “A distance of 100 mm from a fixed caudal point (not visible in figure) is marked cranially (black line).” Or provide a better figure.
  1. I am not sure whether the patients were in Trendelenburg during baseline measurement of if they were placed in Trendelenburg after baseline measurement. If the latter is the case, I would suspect that the measuring tape would move a few millimeters from baseline just from the action of placing the patient in Trendelenburg which would prove the measurement tool to be unstable.
  • Patients were positioned prior to the initial measurements (line 149).
  • From line 22-25, 148-155, 214-216 and the Table 4 legend I gather, that the following methodology was applied during the present trial:
    • Patient is anesthetized and laryngeal mask placed à
    • Patient is placed in Trendelenburg position à
    • A rocuronium dose leading to an NMB degree of PCT=0 is administered à
    • Intubation à
    • The baseline 100 mm mark is made on the abdomen à
    • Capnoperitoneum is established à
    • Surgery begins à
    • The new distance from the 100 mm mark is measured during the five degrees of NMB during surgery à
    • Surgery ends à
    • Extubation
  • Is this correct? The sequence of events should be completely clear in the method section, and additional information from the result section should not be necessary to understand the methodology. Please clarify these details in section 2.4 of the manuscript and consider adding them to the timeline.
  • In line 214-216, the authors write: “After implementation of the capnoperitoneum, abdominal wall length increased about 10 -13 mm (10-13%), compared to baseline”. How did the authors reach this range result of 10-13 mm increase? Is it a range compiled from measurements during all five degrees of NMB? This is unclear and not reflected in Table 4.

  1. Results
  2. I generally struggle with interpretation of the provided tables. The symbols associated with the p-values refer to more than one observation. It should be completely clear which analyses each p-value refers to.
  • The p-values have been rearranged.

Some issues regarding the tables remain:

  • Table 2
    • Please consider the grammar of the legend. I am not a native English speaker, but the commas seem incorrect and confusing to me.
    • In line 246-247, the authors write: “Additionally, the required doses of anesthetics and analgesics did not differ between the time points of measurements.” When looking at Table 2 for statistical support of this statement, it cannot be found. No measures of statistical significance (such a p-values) are provided in the table, and it remains unclear whether or not there was a difference in doses and numbers of boli between the five time points. In my opinion, this renders the table less important.
    • How am I to interpret the number of boli given? What does it tell the reader that 12 boli of sufentanil were given during an NMB of degree PCT=0 until the next degree of NMB was reached?
    • What was the “next degree of the neuromuscular block was reached” (legend table 2) after full recovery (TOFR>90%)?

  • Table 3
    • I am not sure what “HR (1/min)” I believe HR is self-explanatory and doesn’t require a unit.
  • Table 4
    • Pease provide and explanation for the TOF abbreviation in the legend.

  1. Discussion
  2. I don’t believe that I understand line 245-246. It needs rephrasing.
  • The paragraph has been rephrased according to the recommendations of reviewer 2 and 3 (line 287-293).
  • The paragraph in line 283-286 should be removed. This paragraph attempts to conclude an association between a higher degree of NMB and better operating conditions. It is based on only six observations, and it is not supported by the data of 50 patients presented in table 5. This is an example of selective reporting.
  1. I am wondering whether surgeons might be biased in their assessment of SRS. If a surgeon requests an increase in NMB because of SRS 1-3 and is then asked to assess SRS a few minutes later (according to the 15 mins SRS assessment algorithm), the surgeon might be inclined to rate the second SRS as higher than it really is, because he/she expects an improved SRS because he/she just requested additional dose of rocuronium.
  • We agree with the reviewer, that this is a methodological problem of the study. The injection of a placebo instead of rocuronium would have been a possibility to minimise this problem. However, this was an observational study, and therefore, we did not want to compromise the results of the surgery by this measure.
  • Would the authors please add this consideration to the discussion? This poses a potential systematic error that could strengthen the association between the quality of operating conditions and the amount of rocuronium administered/degree of NMB. This should be included in the discussion, especially as a point of concern for future researchers that would consider executing a similar trial.
  1. In line 228-229, the authors write: “Moreover, another result of clinical relevance was, that an intense NMB did not improve surgical conditions compared to a moderate block during gynecologic laparoscopic surgery.” Could the authors please discuss if this a surprising conclusion; whether or not it complies with current literature; and why an intense NMB – contrary to what one might think – didn’t improve surgical conditions?
  2. In line 265-266, the authors write: “This result leads to the question, if the measurement of the abdominal wall length generally is a suitable tool in order to assess the NMB adequately…” I wonder why this way of assessing NMB proved disappointing. Would the authors please add their theories as to why this might be?

REFERENCES

  1. Schuller PJ, Newell S, Strickland PA, Barry JJ. Response of bispectral index to neuromuscular block in awake volunteers. Br J Anaesth. 2015;115 Suppl 1:i95-i103.

Reviewer 2 Report

Soltesz and colleagues submitted a revised version of their manuscript. The raised concerns were adequately addressed. I have a few remaining remarks:

Title: please insert a hyphen or start a new line before “A prospective trial”. Or maybe even omit.

2.2 Anesthesia: did you measure arterial carbon dioxide? I assume you ventilated the lungs to end-tidal carbon dioxide values.

Discussion:

- line 259: You would have to cite POPULAR (Kirmeier et al.) at this point. Your citation 25 is an explanatory analysis showing that at a cut-off of TOFR 0.95 risk for pulmonary complications is not increased anymore. Information which you could also include, if you like.

-Please delete the last sentence: “Based on the data…. or even avoided intraoperatively” You (unfortunately!) have no data to support this.

Conclusion: Change “Generally, an” to “In our study”. Final sentence: not minor clinical relevance, but no clinical relevance! Suggestion: Conclude: “In gynecologic laparoscopic operations, intense NMB neither increases abdominal wall length, nor the quality of surgical operating conditions.”

Author Response

This manuscript is a resubmission of an earlier submission. The following is a list of the peer review reports and author responses from that submission.

Round 1

Reviewer 1 Report

This article is about a prospective, non-blinded study to assess the Influence of neuromuscular block (NMB) on abdominal distension and operating conditions during gynecologic laparoscopic operations.

The study is very interesting, but this article currently has the following critical issues including the study design and method, sample size calculation, interpretation of the results and several technical issues, which should be properly answered and revised.

* Generally in the abstract and main text

An English language editing including grammar, spacing and punctuation marks is required for the paper.

The paper should be more strictly followed by the instructions for authors (for example, when abbreviations are used, full expression of the abbreviations following the abbreviated word in parentheses should be given at the first use. Thereafter, only the abbreviation should be used instead of the full expression, e.g. neuromuscular block (NMB), post tetanic count (PTC).

The degree of the NMB in the study (page 2, lines 79-81) was not classified by the standard criteria of the degree of NMB. I wonder why the 5 stages were classified with the criteria in the study (extreme NMB: PTC = 0; deep NMB: TOF = 0 and PTC = 1–5; moderate NMB: PTC > 5 and TOF = 0–1; recovery: TOF > 1; complete recovery: TOFR > 90%.). Please present the reason.

The standard terms and criteria of the degree of NMB are as follows: intense (profound) NMB: TOF = 0 and PTC = 0; deep NMB: TOF = 0 and PTC ≥ 1; moderate NMB: TOF 1-3; light (shallow) NMB: TOF 4 (TOF ratio 0.1-0.4); minimal NMB: TOF ratio >0.4, but <0.9; full recovery: TOF ratio ≥0.9 (ref: Clinical Anesthesia 8E - Paul G. Barash, et al. LWW, 2017, p 1401, Table 21-12). At least, the NMB terms for the 5 stages used in the study should be properly edited by this standard criteria and terms.

I do not understand why the study design was non-blinded. A blinded assessment of abdominal wall length and SRS seems to be possible.

This study has several potential risks of selective outcome reporting. For example, there is no description about a criteria or dosage for sufentanil or rocuronium bolus injection, and blood pressure and heart rate measurement in the experimental section. More importantly, a subgroup analysis of patients with a BMI > 24.9 or < 25 can be a typical bias of selective outcome reporting, which has a higher risk of arbitrary interpretation about the study results. It is very problematic. The criteria of BMI, itself seems to be meaningless.

The description in the statistical analysis is poor.

* Abstract

Page 1, lines 18-19: ~ at 5 stages: Posttetanic Count=0; Posttetanic Count=1-5; Train of Four Count =0-1; Train of Four Count >1; Train of Four Ratio>90%. -> The 5 stages of the NMB should be defined with the terms of NMB degree as well as the monitoring measurements: e.g. posttetanic count (PTC) = 0 (intense NMB). Please present the number of total patients included in the analysis. Also, the results of the abdominal wall length between the different stages in all patients are more important than them of a subgroup and thus they should be presented instead of the results of a subgroup.

* Introduction

The hypothesis of the study should be presented.

* Experimental section

Page 2, line 53: American Society of Anesthesiologists Physical Status ASA 1-2 -> American Society of Anesthesiologists (ASA) Physical Status 1-2

Page 2, line 61, line 89: How were propofol and remifentanil infused? (manually or using target controlled infusion [TCI] devices) If TCI devices were used, their types, the model for drugs and their concentration should be properly presented in detail.

Page 2, line 69: After induction of anesthesia, the acceleromyograph was calibrated over a period of 10–15 min until a stable signal without drift could be obtained. -> After induction of anesthesia, the acceleromyograph was calibrated and stabilized over a period of 10–15 min until a stable signal without drift could be obtained.

Page 2, line 71: Afterwards, the degree of the neuromuscular block was assessed every 15 s by acceleromyography. -> Afterwards, the degree of the neuromuscular block was assessed using TOF stimulation every 15 s by acceleromyography.

Page 2, lines 79-81: The degree of the NMB in the study was not classified by the standard criteria of the degree of NMB. I wonder why the 5 stages were classified with the criteria in the study (extreme NMB: PTC = 0; deep NMB: TOF = 0 and PTC = 1–5; moderate NMB: PTC > 5 and TOF = 0–1; recovery: TOF > 1; complete recovery: TOFR > 90%.). Please present the reason.

The standard terms and criteria of the degree of NMB are as follows: intense (profound) NMB: TOF = 0 and PTC = 0; deep NMB: TOF = 0 and PTC ≥ 1; moderate NMB: TOF 1-3; light (shallow) NMB: TOF 4 (TOF ratio 0.1-0.4); minimal NMB: TOF ratio >0.4, but <0.9; full recovery: TOF ratio ≥0.9 (ref: Clinical Anesthesia 8E - Paul G. Barash, et al. (LWW, 2017) p 1401, Table 21-12). At least, the NMB terms for the 5 stages used in the study should be properly edited by this standard criteria and terms.

Page 3, in Figure 1, indicate the caudal and cranial directions.

Present a method of NMB reversal and patient management at a recovery period including extubation, and/or postoperative care and adverse events.

In the statistical analysis, the authors should present each statistical method used for the comparison of all variables including patient’s characteristics and clinical data in detail as well as main outcomes. Which statistical program was used for the analysis?

In the sample size calculation, I wonder whether “a difference of 10 mm in increase of abdominal wall length” means a difference between which time points. Please clarify it. In addition, did not you consider a drop rate? Why did you include 60 patients, not 45 patients for the study?

* Results

Page 4, line 151, Thirteen patients required additional doses of rocuronium to reach a posttetanic count of 0. -> In the Table 1, n=12. Please check it. Page 4, line 153: 44 cumulative additional sufentanil bolus injections were performed. -> I do not understand this sentence. In the Table 3, RR syst (mm Hg) -> What did RR mean? In the Table 4, P value should be presented by numeric value, not n.s.

Reviewer 3 Report

Soltesz and colleagues present a study in which the surgical rating score as well as a new tool for measuring abdominal distention were assessed during different degrees of neuromuscular block during laparoscopic surgery with a moderate pressure capnoperitoneum at 12 mmHg. Overall in all patients, there was no difference in SRS values or differences in abdominal wall length at the different depths of neuromuscular block. Only in obese patients the increase in abdominal wall length was significant when PTC = 0 was compared to TOFR>0.9. And, in comparison with patients with a BMI<25, patients with a BMI > 24.9 had slightly lower SRS. Irrespective of this, the average values showed that surgeons had “good” to “optimal” operating conditions in all patients. Unfortunately, no information about any other intraoperative (e.g. adverse events, pressure alarms capnoperitoneum or ventilation etc.) or clinical outcome parameter (e.g. pain, nausea etc.) is given.

The topic of this study is hotly debated in operating rooms around the world. The information presented in this study is, therefore, of importance. However, I strongly urge the authors to change the message that needs to be brought across.

Regarding influence of NMB on surgical conditions: this question is not new and there have been a couple studies published on this topic. The authors decided to use the 5-point surgical rating scale (SRS) which I find rather crude. However, since 2 authors are from the Ob/gyn department and the scores were all between “good” and “optimal”, the surgical conditions and the evaluation thereof, indeed seemed to be adequate. This is the first key message: NMB or noNMB – the surgeons still have good to optimal operating conditions in all patients (even the ones with a BMI >25). The second key message: the SRS is not different between TOFR>0.9 and PTC=0 – this means NMB cannot improve surgical conditions.

Regarding the new technique to measure abdominal wall distension using a measuring tape longitudinally taped to the abdominal wall: what one could have intuitively assumed is that the deeper the neuromuscular block the more the abdominal wall is distended. However, this is not the case. I do not think that measuring the distention of the abdominal wall will become of clinical relevance. However, the insufflation of carbondioxide to a moderate intraabdominal pressure of 12 mmHg distends the abdomen irrespective if the patient has a neuromuscular block or not. This is, as I wrote before, what I would focus on: no matter how deep or shallow neuromuscular block is, if anesthesia is adequate, NMB has no influence on abdominal wall distention or operating conditions. In daily practice anesthetists are often confronted with a surgeon asking for more or deeper NMB. With the results of this study in combination with the results of the ESA-CTN study POPULAR, which showed that muscle relaxants increase the incidence of post-operative pulmonary complications, anesthetists should now argue that further (intraoperative) NMB will a) not improve surgical conditions and b) could even harm the patient.

NMB makes sense for induction if the trachea of the patient is intubated (better intubating conditions, less intubating injury – see Combes X, BJA 2007 and Mencke T. Anesthesiology 2003) and it also makes sense for establishing capnoperitoneum. Afterwards however, the authors of this study should argue there is no benefit (with the exception of certain safety aspects – please see Blobner M: NISCO Study, Surg. Endosc. 2015) and therefore undifferentiated injection, simply because the surgeon thinks surgical conditions could be improve, should be avoided. During the time course of the operation, the patient can recover from NMB, as there is no difference in surgical conditions along the complete recovery curve from NMB. The potential threat to the patients at the end of the operation /anesthesia regarding post-operative pulmonary complications can be averted if the patient recovers to a TOFR > 0.95 (Blobner BJA 2020). This can be achieved either spontaneously or with sugammadex or neostigmine reversal (Kaufhold BJA 2016, Schaller Anesthesiology 2010). With this concept you have the benefit for the patient, the anesthetist and the surgeon at the beginning of the procedure, no unnecessary drug application during the operation and no drug-induced harm to the patient at the end.

Specific comments:

Title: I would completely change the title of the manuscript to bring the message across in a better way. Suggestion: “Depth of neuromuscular block does not influence abdominal wall distention or surgical conditions during gynecologic laparoscopic operations.”

Abstract:

Line 26: “Generally, a neuromuscular block increased abdominal wall length…” I cannot find any data to support this conclusion. Overall, NMB had no influence on abdominal wall distention in this study. The capnoperitoneum per se distended the abdominal wall.

Materials and Methods:

What kind of laparoscopic procedures were performed? How was acceleromyography calibrated? Please change the term “extreme” to “intense” according to the GCRP-Guidelines (Acta Anaesthesiol Scand 2007;51:789-808). Please also change this in the abstract. Also, please change recovery (page 2, line 80 to shallow) The span between TOF>1 and TOFR>0.9 is huge. When exactly were patients of the group TOF>1 assessed (e.g. TOFR 0.85 is TOF>1 but < TOFR 0.9)? The general anesthetic regimen appears to be a rather “light anesthesia” with 3-4 mg/kg/h Propofol and remifentanil at 0.12-0.14 µg/kg/min (even though it is BIS controlled). This necessitated the additional boluses of sufentanil. Why did the authors choose to use two different opioids rather than only using remifentanil and therefore possibly being able to see an effect of the opioid? Did the patients have a capnoperitoneum or pneumoperitoneum? Please be consistent throughout the text. Do you have any other outcome parameters such as pain scores, adverse events etc. Omit the differentiation between patients above and below 25. The study was not powered for this. 

Results:

Table 3: In the text you write HR is higher at TOF 0-1 compared to PTC = 0. In the table it says TOF 0-1 compared to TOFR>90%.

Discussion:

Line 186: I cannot find data to support the statement first line of the discussion. Only in patients with a BMI >24.9 was there a slight effect. Line 188-190: also disagree that the new methods of abdominal wall distention is able to distinguish between different degrees of NMB – this study did not show this. Irrespective of this, I do not think that measuring abdominal wall distension will ever become clinical practice to monitor or guide application of muscle relaxants.